# Difficulties in the Diagnostics and Treatment of Hashimoto’s Encephalopathy—A Systematic and Critical Review

**DOI:** 10.3390/ijms25137101

**Published:** 2024-06-28

**Authors:** Nikola Pempera, Miłosz Miedziaszczyk, Katarzyna Lacka

**Affiliations:** 1Students’ Scientific Society, Endocrinology Section at Department of Endocrinology, Metabolism and Internal Medicine, Poznan University of Medical Sciences, 60-355 Poznan, Poland; nikolapempera@gmail.com; 2Department of General and Transplant Surgery, Poznan University of Medical Sciences, 60-355 Poznan, Poland; m.miedziaszczyk@wp.pl; 3Department of Clinical Pharmacy and Biopharmacy, Poznan University of Medical Sciences, 60-806 Poznan, Poland; 4Department of Endocrinology, Metabolism and Internal Medicine, Poznan University of Medical Sciences, 60-355 Poznan, Poland

**Keywords:** Hashimoto’s thyroiditis, encephalopathy, steroidotheraphy, steroid-responsive encephalopathy associated with autoimmune thyroiditis, anti-TPO

## Abstract

Hashimoto’s encephalopathy (HE) has been a poorly understood disease. It has been described in all age group, yet, there is no specific HE marker. Additionally, the treatment data in the available studies are frequently divergent and contradictory. Therefore, the aim of our systematic and critical review is to evaluate the diagnosis and treatment of HE in view of the latest findings. The databases browsed comprised PubMed, Scopus, and Google Scholar as well as Cochrane Library, and the search strategy included controlled vocabulary and keywords. A total of 2443 manuscripts were found, published since the beginning of HE research until February 2024. In order to determine validity of the data collected from studies, bias assessment was performed using RoB 2 tool. Ultimately, six studies were included in our study. HE should be considered in the differential diagnosis in patients with psychiatric and neurological symptoms. According to our findings, negative thyroid peroxidase antibodies (anti-TPOs) may represent a valuable parameter in ruling out HE. Nonetheless, this result cannot be used to confirm HE. Furthermore, the proposed anti NH2-terminal-α-enolase (anti-NAE) is non-specific for HE. The effectiveness of glucocorticoid therapy is 60.94%, although relapse occurs in 31.67% of patients following the treatment. Our review emphasizes the significance of conducting further large-scale research and the need to take into account the potential genetic factor.

## 1. Introduction

Hashimoto’s encephalopathy (HE) remains a disease that is poorly understood. The first case of HE was described by Lord Brain in 1966. The patient was a 49-year-old man who presented with recurrent cognitive disorders, impaired mental status, hallucinations and stroke-like events [1]. So far, the sources have described over 300 cases of HE, with approximately 30 occurring in children [2]. 

HE is one of the most elusive diseases, and its etiology as well as treatment still need to be clearly defined [3]. Many researchers suggest that the reported HE cases have not been properly diagnosed, whereas others claim that HE constitutes modern neuromythology [4]. HE is a very rare disease, with an estimated prevalence of approximately 2.1/100,000 in the adult population [5]. Although it has been described in all age groups (including children), it is most common in individuals between 40 and 60 years of age. In addition, it occurs much more frequently in women than in men (4:1), which may result from the higher incidence of Hashimoto’s disease in the female population [5,6]. Furthermore, HE co-occurs with other autoimmune diseases (such as Sjögren’s syndrome, type I diabetes, and systemic lupus erythematosus) in approximately 30% of patients [7]. Notably, it is also highly likely that HE remains underdiagnosed [8,9].

Despite over 55 years of observation, the pathogenesis of HE is still unclear [3], albeit it has been suggested that apoptosis may be one of the underlying mechanisms. Programmed cell death is essential for the maintenance of homeostasis; however, when it is not regulated, it may contribute to disorders, such as HE [10,11]. Moreover, the presence of antithyroid antibodies, a good response to glucocorticosteroids and the occurrence of exacerbation and remission episodes support the autoimmune character of the disease. Nevertheless, up to the present date, no correlation has been found between the antibody level and the severity of symptoms [1,12]. Despite the high concentration of antithyroid antibodies in serum and cerebrospinal fluid, they have not been found in nervous system structures in the conducted autopsy and experimental studies [6,8]. In fact, in HE patients, no decrease in antithyroid antibodies was observed following steroid treatment [8,9]. Yet, a positive correlation was found between the level of thyroid peroxidase antibodies (anti-TPOs) and thyroid fibrosis. Therefore, the criterion of high antithyroid antibodies in the diagnosis of HE remains controversial [13]. It is worth bearing in mind that the available scientific literature provides no clear diagnosis or treatment of Hashimoto’s encephalopathy. This, in turn, emphasizes the need to summarize the existing evidence and to identify questions that need to be addressed as soon as possible. Therefore, the aim of our systematic and critical review is to evaluate the diagnosis and treatment of HE in the light of the latest developments in this area.

## 2. Materials and Methods

### 2.1. Search Strategy

This review is designed following the PRISMA 2020 guidelines (Appendix A). We performed a PubMed, Scopus, Cochrane Library and Google Scholar search throughout February 2024. Search terms, including “Hashimoto’s encephalopathy”, “steroid responsive encephalopathy associated with autoimmune thyroiditis”, “SREAT”, “anti-TPO” and “thyroid peroxidase antibodies” have been used as isolated keywords, as well as keywords used in combination. The first selection of studies was based on the title and keywords. Overall, 2443 papers were found, published since HE research commenced until February 2024, since even the oldest articles were significant for obtaining as many data as possible. The country of origin and form of the publications were irrelevant. Two analysts, working independently, verified the search engine results to select the most relevant studies. In total, the presented systematic review comprised 6 studies and the study protocol was registered (ID CRD42024525927).

### 2.2. Data Extraction

The selection strategy involved the selection of studies for general information about Hashimoto’s encephalopathy. As a result, 2443 records were identified and screened independently. Each analysts prepared their own list of records identified as relevant to the study. Subsequently, 2437 records were excluded due to the type of paper. The exclusion criteria comprised meta-analyses, case reports, reviews, only abstract available, papers in languages other than English, and duplicates. Next, these record lists were double-checked by both analysts, and 6 full-text records were assessed for eligibility. Inadequate information, as well as the presence of other encephalopathies, constituted the exclusion criteria. Moreover, only full-text manuscripts were included. Both analysts read the papers independently and duplicates were removed. The methodology of identifying the original papers is demonstrated in Figure 1.

### 2.3. Qualitative Analysis and Synthesis

In order to determine the validity of the data collected from studies, bias assessment was performed. It was conducted using the RoB 2 tool for each individual study, where two researchers addressed signaling questions to determine the risk of bias. The sources obtained for bias assessment included papers and study protocols, if available. The effect of bias assessment has been presented as a forest plot in Figure 2. In general, the risk of bias is used to determine the validity of the data extracted from the studies that met the inclusion criteria, and it is essential for drawing conclusions from the data by assessing the risk of bias. Studies where the risk was identified as ‘low’ serve as the basis for drawing the final conclusions. Six papers were found to be the most relevant to the study, and thus they were included in this systematic review. Statistical calculations were performed using the direct method of arithmetic mean of grouped data.

## 3. Results

### 3.1. Pathogenesis of Hashimoto’s Encephalopathy

The pathogenesis of Hashimoto’s encephalopathy remains unclear [3,9], although thyroid dysfunction appears not to play a role. HE is fundamentally considered to be an autoimmune disease [6,9]. Moreover, no correlation has been found between thyroid antibody concentration, the severity of the disease and the constellation of symptoms [20]. Despite high levels of antithyroid antibodies in the serum and cerebrospinal fluid, they have not been observed in the central nervous system structures in autopsy studies [6,8]. 

Therefore, it would be relevant to address the issue of what induces and affects the development of HE. Essentially, three major mechanisms have emerged [14,21,22,23,24,25,26,27,28,29,30]. The first one is related to vasculitis. The concept assumes that the brain microvasculature is altered and this condition leads to edema, or that vasculitis is a result of endothelial inflammation [21]. Histopathological and morphological examinations in patients with HE revealed the presence of chronic, limited inflammatory lesions of the cortex and meninges, which have been referred to as NAIM (nonvasculitis autoimmune inflammatory meningoencephalitis) [22]. A paper published by Nolte et al. describes an association between HE and lymphocytic infiltration or inflammatory lesions of the brainstem and gray matter due to vasculitis [23]. The second hypothesis is based on the presence of anti-thyroid antibodies. However, as mentioned above, no relationship has been found between the increase in the antibody level and the severity of Hashimoto’s encephalopathy [14,25,26]. In recent years, particular attention has been paid to the aggressive form of HE associated with elevated IgG4 levels. It occurs more often in the male population (5:1), begins at a younger age, and is characterized by higher levels of antithyroid antibodies [27,28]. It has been observed that IgG4 antibodies are present in 70% of patients with hypothyroidism. When it comes to euthyroid patients, only 4% present high IgG4 concentration [29]. Hosoi et al. described a 60-year-old patient with severe HE and elevated IgG4 concentrations in serum and cerebrospinal fluid. The patient’s IgG4 index was than lower that of IgG, indicating passive transport of IgG4 across the blood–brain barrier. Moreover, after corticosteroid treatment, serum IgG4 levels decreased, but were indeterminate in the cerebrospinal fluid. The described case may highlight the suspected influence of the IgG4 fraction on the development of neurological abnormalities, including HE [30]. The third hypothesis stems from toxic effects of the thyrotropin-releasing hormone (TRH). Nevertheless, only one study has been conducted which demonstrated that TRH triggered myoclonus and tremor similar to the symptoms that patients experience during exacerbation of HE symptoms [24]. The hypothesis of a direct toxic effect of TRH (thyrotropin-releasing hormones) on nerve cells, inflammatory lesions of the brain and medulla, in the course of demyelination, was also considered as a causative factor when a patient developed myoclonus and tremor after an intravenous TRH infusion [24]. However, only one study pertaining to this issue has been conducted up to the present date [24]. 

It is worth bearing in mind that some of the studies included subjects of just one nationality, mostly Asian (five out of six presented studies). Consequently, the question arises as to whether there is a gene responsible for the development and the course of this disease. Therefore, since there are no studies indicating a genetic basis for HE, it is our belief that conducting such research is essential. 

We present the mentioned suspected mechanisms in Figure 3.

### 3.2. Diagnostic Criteria of Hashimoto’s Encephalopathy

It is vital to note that, to date, no clear diagnosis or treatment has been established for Hashimoto’s encephalopathy, and the diagnosis itself is by exclusion. Therefore, firstly, it is essential to exclude all potential disorders of the central nervous system. Yet, in the available studies, different authors have used various criteria in the course of the diagnostic process of HE. We present and compare the criteria in Table 1.

Kishitani et al., in their research, relied merely on the presence of antithyroid antibodies and the responsiveness of immunotherapy. Their study involved 14 patients with H,E and 26 individuals constituted the control group (consisting of 12 healthy subjects and 14 participants with Hashimoto’s thyroiditis who did not present with any neuropsychiatric symptoms). In three patients from the study group, a history of Hashimoto’s disease was present prior to the suspicion of HE. The limitations of the abovementioned study include a small number of patients and the inclusion of only Japanese subjects.

In turn, Mamoudjy et al. based their diagnosis on the clinical signs of hypothyroidism (goiter, fatigue and/or stunted growth) and high serum anti-TPO antibody levels. The researchers compared the group of eight children to thirty-four controls, and found that four of eight children had a personal or familiar history of autoimmune disease. 

Tang et al., in their study, applied the extended criteria for HE, which included encephalopathy manifested by clouding of consciousness, cognitive impairment, seizures or neuropsychiatric features, and elevated anti-TPO antibodies with euthyroid status (serum-sensitive thyroid-stimulating hormone [TSH], 0.35–5.5 uIU/mL). The criteria they applied also encompassed no alternative infectious, toxic, metabolic, vascular or neoplastic etiology related to the neurological symptoms observed in blood, urine, cerebrospinal fluid (CSF) or neuroimaging examinations. The researchers considered 13 patients, although they did not investigate the control group, which constitutes the most significant limitation of this study. Furthermore, all of the participants were Chinese. According to their medical history, five subjects presented with autoimmune diseases and three with thyroid disorders. Moreover, one patient reported systemic lupus erythematosus and another suffered from Raynaud’s syndrome. 

In another study, Manjunatha Suryanarayana Sharma et al. applied the following diagnostic criteria: acute or subacute onset of altered mental status (AMS), elevated antithyroid antibodies, and rapid response to the administration of corticosteroids reflected in patients’ mental status, as well as absence of structural, infectious or other metabolic factors, which could account for the AMS and its response to steroids. Their study comprised 13 participants, yet, they also failed to report the control groups. In five patients, a history of hypothyroidism preceded the onset of neurological symptoms, and one presented with euthyroid goiter. 

The study by Dumrikarnlert et al. comprised the following HE criteria: encephalopathy with seizures, myoclonus, and hallucinations, or stroke-like episodes, as well as thyroid disease (subclinical or mild overt). Their criteria involved also brain MRI—normal or with nonspecific abnormalities—the presence of serum thyroid antibodies, the absence of other neuronal antibodies in the serum or CSF, and the exclusion of alternative causes of encephalopathy by differential diagnosis. The study group consisted of 13 patients, whereas the control group involved 91 patients. 

Interestingly, Mattozzi et al. employed similar criteria as Dumrikarnlert et al. in their study. The criteria included subacute onset of cognitive impairment, psychiatric symptoms or seizures, euthyroid status or mild hypothyroidism, serum thyroid peroxidase antibodies (TPOAbs) > 200 IU/mL, absent neuronal antibodies in the serum/CSF and no other etiologies. The study group consisted of 24 individuals and the control group of 13.

### 3.3. Laboratory Findings

#### 3.3.1. Family History of Autoimmune Diseases 

Out of 48 interviewed patients, 18 (38%) reported a history of autoimmune disease in their family or personal history. Hence, HE appears highly likely to co-occur with other autoimmune diseases. However, no clear data are available with regard to the most common comorbidities. The existence of an autoimmune disease in the family or in the patient should indicate the need to monitor the disease in order to limit its development and to start treatment as early as possible. Laboratory results obtained from the sources are presented in Table 2.

#### 3.3.2. Thyroid Parameters

In the available scientific literature [14,15,16,17,18,19], anti-TPO was measured in 71 patients, and was elevated in 67 (94%). According to the researchers, an increased level of anti-TPO was considered as a criterion for HE. Nevertheless, these antibodies are not specific to this disease. In fact, Mamoudjy et al. reported that even the median level was 4043.3 ± 2969.8 IU/mL. In all the reviewed studies, abnormal TSH level was observed in 7/34 (21%) cases, and abnormal fT4 level in 2/22 (9%). Additionally, there is no clear evidence if the patients were euthyroid patients, due to the treatment with levothyroxine, or without any medications. Table 3 shows the results regarding the sensitivity and specificity of thyroid peroxidase (anti-TPO) as a diagnostic parameter of Hashimoto’s encephalopathy.

It should be emphasized that, according to the sources, the sensitivity of anti-TPO is 85.714%, whereas the specificity is 60.563%. Therefore, it is possible to conclude that the positive predictive value is 46.154%, and the negative predictive value is 91.489%. This, in turn, demonstrates that anti-TPO may constitute an extremely useful parameter, indicating that a person with a negative result does not actually suffer from HE. However, anti-TPO level cannot be used to state conclusively that a patient with a positive result has HE.

#### 3.3.3. Anti NH2-Terminal-α-Enolase (Anti-NAE) Pre-Treatment

As it seemed essential to find a specific parameter specifically characterizing HE, researchers suggested antibodies directed against α-enolase. In their study, Kishitani et al. measured the concentration of anti-NAE antibodies in 14 patients, and the results obtained were between 320 IU/mL and 40.960 IU/mL, which shows that the proposed parameter could be considered specific. In contrast, however, Mattozzi et al. demonstrated an elevated level in just one patient out of the suspected twenty-four HE cases, and in one control. This, in turn, shows that anti-NAE is not a specific diagnostic indicator in all the patients. It is highly likely that, due to different diagnostic criteria, Kishitani et al. and Mattozzi et al. received diverse results related to the specificity of anti-NAE. This highlights the particular need for a further deeper understanding of the disease and a search for new diagnostic parameters.

#### 3.3.4. Other Laboratory Findings

Hyponatremia was reported in 6 of 14 patients in the study by Kishitani et al. Out of all the studies, 42 of 84 (50%) subjects presented elevated protein levels in cerebrospinal fluid (CSF). Furthermore, 53 of 70 patients presented abnormalities in electroencephalography (EEG), where generalized slowing was noted in most patients with abnormal EEG. Aminotransferase level was elevated in 8 of 26 patients (31%) and anti-nuclear antibodies (ANA) were elevated in 8 of 26 participants (31%). Rheumatoid factor was positive in 3 of 26 (12%) subjects. Additionally, the researchers observed elevated C-reactive protein (CRP) in 5 of 39 patients (13%). It is of note that the abovementioned parameters do not indicate the presence of characteristic HE changes in the parameters. In fact, they suggest that Hashimoto’s encephalopathy is a diverse disease, which differs in various patients.

### 3.4. Clinical Manifestation of HE according to the Analyzed Studies

Patients with Hashimoto’s encephalopathy present with various clinical manifestations. The most frequently reported are disturbances of consciousness, impaired memory and sleep disturbance. These symptoms are found in numerous neurological and psychiatric disorders, which makes HE diagnosis even more challenging. Patients are often misdiagnosed and need to be subsequently re-diagnosed. The summarized frequency of symptoms is presented in Figure 4.

The issues related to misdiagnosis prolong the waiting time for the optimal treatment. Our analysis summarizes those reported in six reviewed original papers (Kishitani et al. [14], Mamoudjy et al. [15], Tang et al. [16], Manjunatha Suryanarayana Sharma et al. [17], Dumrikarnlert et al. [18] and Mattozzi et al. [19]), so as it is possible to draw conclusions regarding their frequency among HE patients. 

Kishitani et al. reported disorders of consciousness in 10 of 14 patients, memory dysfunction in 9 of 14 patients, psychiatric symptoms in 7 of 14 patients, seizures in 6 of 14 patients, involuntary movement in 2 of 14 patients and respiratory impairment in 1 of 14 patients.

Mamoudjy et al., in turn, observed disorders of consciousness in seven of eight patients, seizures in four of eight patients and behavioral changes in four of eight patients.

In another study, Dumrikarnlert et al. found disorders of consciousness in 6 of 13 patients, psychiatric symptoms in 6 of 13 patients, sleep disorders in 5 of 13 patients, seizures in 5 of 13 patients, behavioral changes in 4 of 13 patients and involuntary movement in 3 of 13 patients.

Mattozzi et al. reported seizures in 12 of 24 patients, psychiatric symptoms in 6 of 24 patients, disorders of consciousness in 6 of 24 patients, memory impairment in 4 of 24 patients, behavioral changes in 4 of 24 patients and involuntary movement in 4 of 24 patients.

Tang et al. found memory disorders in 7 of 13 patients, psychiatric symptoms in 5 of 13 patients, sleep disturbance in 4 of 13 patients, seizures in 4 of 13 patients, seizures in 4 of 13 patients and involuntary movement in 1 of 13 patients.

According to Manjunatha Suryanarayana Sharma et al., memory disturbance was recorded in 10 of 14 patients, behavioral changes in 10 of 13 patients, sleep disturbance in 9 of 13 patients, seizures in 6 of 13 patients, psychiatric symptoms in 5 of 13 patients and involuntary movement in 4 of 13 patients.

### 3.5. Treatment of HE According to the Sources

Table 4 shows the current treatment options for Hashimoto's encephalopathy.

## 4. Discussion

### 4.1. NH2-α-Enolase—Is It Specific?

Interestingly, a new antigen, α-enolase, has recently been discovered in the brains of HE patients. In addition, high levels of α-enolase antibodies have been found in the serum and cerebrospinal fluid of the same patient group [31]. In fact, it was even considered a potential biomarker of Hashimoto’s encephalopathy [25]. However, it subsequently turned out that α-enolase was not specific to the blood vessels of the brain, and that it appeared in most cells of the human body [32]. It was demonstrated that increased values of α- and γ-enolases also occur in other autoimmune diseases, such as rheumatoid arthritis, systemic lupus erythematosus, antineutrophil cytoplasmic antibodies (ANCAs) positive vasculitis, autoimmune nephritis, primary Sjögren’s syndrome, systemic sclerosis, primary biliary cirrhosis, inflammatory bowel disease and celiac disease [32,33,34]. Furthermore, one of the conducted studies has shown that the enolases have also been identified in a patient with Creutzfeldt–Jakob disease, as well as in patients with limbic encephalitis [14]. Anti-NAEs are expressed in endothelial cells, which accounts for their presence in other vasculitic diseases, such as Kawasaki disease [35]. It is suggested that anti-NAE should be considered as a potential risk-modifying factor [36]. This, in turn, prompted the researchers to hypothesize that HE may also constitute a form of immunopathological vasculitis [20,21]. Nonetheless, in the study by Mattozzi et al., in a group of 24 people, anti-NAEs were detected in 1 patient with HE and in 1 with another type of encephalopathy, which suggests that these antibodies do not contribute to the diagnosis of HE [29]. 

### 4.2. Symptoms—Two Different Types of HE

The clinical course of HE is not specific. However, cognitive dysfunction has been described as the initial manifestation in over 80% of patients, and behavioral and personality disorders in over 90–100% [5,7]. The disorders observed in the course of Hashimoto’s encephalopathy vary. Some patients develop tremor, myoclonus, ataxia, stupor or coma [37,38]. Patients less frequently report persistent headaches, gait disturbances, isolated nystagmus, opsoclonus-myoclonus syndrome, Parkinsonian syndromes, fatigue syndrome, sleep disturbances or hallucinations [5,6]. Seizures occur in approximately 60–70% of HE patients [8,39]. 

According to the conducted reports, HE is divided into two types. In the first type, recurrent incidents appear which suggests that the underlying cause is vascular (stroke-like incidents, corresponding to transient cerebral ischemia), with frequent disorders of cognitive function, although without seizures. It occurs more rarely and is milder, and thus the prognosis indicates higher chances of recovery. The other type is progressive in nature, with epileptic seizures, disturbances of consciousness, psychiatric symptoms such as mania, depression or psychosis, and increasing dementia syndrome. It is more common, usually has acute onset and rapid course, and usually progresses without relapses. The manifestations of the second type seem to be more intense, and hence the prognosis usually remains poorer [2,6,30].

### 4.3. Diagnostics 

According to the diagnostic criteria suggested by Graus et al., the diagnosis of HE is by exclusion [40]. Basic laboratory tests usually show no specific abnormalities, and inflammatory parameters, such as CRP, usually tend to be negative. Patients do not present with fever, and TSH levels are usually normal, hence, no correlation between thyroid function and patients’ clinical condition is observed. In most of the described cases, patients were euthyroid or subclinically hypothyroid, presenting elevated serum levels of anti-TPO (*thyroid peroxidase antibodies)* and/or anti-TG (*thyroglobulin antibodies*) [37,41]. CSF examination revealed elevated protein levels in approximately 85% patients, which normalized following steroid treatment. In rare cases, a slight lymphocytic pleocytosis may also be found. 

In their study, Dumrikarnlert et al. [18] adopted brain MRI as a diagnostic criterion, and revealed non-specific white-matter changes in an area or areas that is/are non-specific to any other symptoms, such as in the corona radiata. None of the lesions were compatible with a diagnosis of multiple sclerosis, neuromyelitis optica, previous stroke, or brain injury [18]. In general, approximately half of MRI findings in HE are normal, whereas the remaining MRI brain imaging show non-specific lesions. In the cases described by Jegatheeswaran et al., the researchers observed and described generalized brain atrophy or an increased signal in the subcortical white-matter area on a T2-weighted/fluid-attenuated inversion recovery (FLAIR) sequence [5,42]. Moreover, the authors emphasized that the changes in MRI did not disappear, despite the clinical improvement following the treatment [42]. Another case involved HE coexisting with sensory ganglionopathy and painful neuralgic amyotrophy [43]. Bearing in mind that the range of the described changes is wide, they are not typical only for HE, and therefore HE may be misdiagnosed as other autoimmune encephalitides. Nonetheless, MRI may be a valuable parameter in the diagnostic process, since it may also indicate other diseases that need to be excluded [42].

According to the studies, the majority of patients show the presence of antithyroid antibodies in the CSF [44]. However, other autoimmune antibodies have also been observed, such as GABAAR (gamma-aminobutyric acid A receptor), NMDAR (N-methyl-D-aspartate receptor), LGi1, Caspr2 (contactin-associated protein 2), *leucine-rich glioma inactivated 1* (LGi1), a-amino-3-hydroxy-5-methylisoxazole-4-propionic acid receptor 1 and 2 (AMPAR1/2) or ANA (antinuclear antibodies) [14,37,39,45]. Approximately 82–98% of patients demonstrate abnormal EEG [8,39]. It is vital to observe that the changes are not specific, and include a decreased generalized basal activity and paroxysmal delta- and theta-wave discharges [46]. Nevertheless, one of the studies found an elevated concentration of antinuclear (ANA) and anti-TPO autoantibodies in 28.3% of patients with idiopathic epilepsy [47]. 

A few case reports also present patients with concomitant peripheral nervous system damage. In all these individuals, the neurological symptoms resolved following steroid treatment [44,48,49]. 

The authors suggest the diagnostic path as presented in Figure 5.

### 4.4. Treatment

There are no clearly established treatment guidelines for HE. The first-choice therapy involves the administration of corticosteroids. Our analysis indicates that they are administered in 81.18% of cases. The drug of choice is most frequently methylprednisolone at a dose of 1 g i.v. for 5 days. However, some physicians emphasize the need for subsequent oral steroid therapy. In that case, prednisone at an initial dose of 50–150 mg daily or 1–2 mg/kg/day is recommended [5,37]. Although the effectiveness of glucocorticoid therapy is 60.94%, relapse occurs in 31.67% of patients after such treatment. If the steroids are ineffective, second-line treatment should comprise methotrexate, azathioprine or cyclophosphamide. In addition, plasmapheresis and immunoglobulin (IVIG) treatments have also been described to be effective, and rituximab has been used successfully in a patient with opsoclonus-myoclonus syndrome [37]. Nevertheless, it is vital to note that seizures may recur, particularly following a reduction in the steroid dose [39,51]. 

### 4.5. Treatment Complications

The treatment results clearly demonstrate that in most cases, steroid therapy significantly improves the patients’ condition. The majority of patients report complete recovery, or observe significant improvement; however, studies have described relapses, and some treated patients require further rounds of steroids and, at times, immunosuppressive therapy with other drugs. Corticosteroids are potentially more effective at the very onset of the disease; therefore, the delay in diagnosis may result in the need for another type of treatment [52]. Interestingly, sources describe cases of spontaneous recovery, as well [22,53]. Long-term steroid treatment is not risk-free, and hence it is worth bearing in mind that it may result in serious side effects. The patients who undergo such a treatment require frequent monitoring of clinical and laboratory parameters. Moreover, side effects may involve most major organ systems, such as musculoskeletal, gastrointestinal, cardiovascular, endocrine, neuropsychiatric, dermatological, ocular, and immunological. All sorts of adverse reactions are possible, including joint necrosis, growth-suppression adrenal insufficiency, osteoporosis, gastrointestinal, hepatic effects, hyperlipidemia or congenital malformations [54].

### 4.6. Steroids in Other Encephalopathies

Encephalopathy is a general term for chronic or permanent brain damage, and it is generally divided into congenital and acquired. Encephalopathy may, for instance, occur as a result of a viral infection. Yet, specific antiviral treatment is available only for a small subset of infections. A number of studies have been conducted which suggest that steroids may be effective in the virus-induced encephalopathies by reducing the secondary inflammation-mediated damage [55]. Moreover, epileptic encephalopathies are conditions in which neurological deterioration predominantly stems from epileptic activity [56]. It has been observed that hormonal treatment with steroids results in an improvement in speech, cognition, behavior and general condition in almost all patients, and it is usually accompanied by an improvement visible on the EEG [57]. The literature shows that encephalopathy may also constitute a serious complication of hypoglycemia in the course of diabetes mellitus. Following the administration of 1 g intravenous methylprednisolone, the 73-year old patient showed a considerable neurological improvement, including even a successful extubation [58]. In contrast, hepatic encephalopathy is a neuropsychiatric complication of cirrhosis or portosystemic shunting. It is evident, based on the abovementioned evidence, that steroids are effectively involved in the treatment of various encephalopathies. Nevertheless, it is necessary to conduct randomized clinical trials to maximize the effectiveness of the therapy. 

### 4.7. Limitations

The main limitation of our analysis involves the inability to calculate the anti-TPO cut-off point due to the lack of data in the analyzed groups. Moreover, it is impossible to compare changes in specific parameters after steroid treatment, since the analyzed sources failed to provide these data. Anti-Tgs, which also play a significant role in pathogenesis of Hashimoto’s disease, were not measured in the studies mentioned. It should be emphasized that, since Hashimoto’s encephalopathy is relatively rare, all analyzed study samples were small. Therefore, further research involving large-group studies is needed.

## 5. Conclusions

The pathogenesis of Hashimoto’s encephalopathy has been unclear. Furthermore, even the diagnostic criteria have not been clearly defined so far, and they comprise encephalopathy with seizures, myoclonus, hallucinations or stroke-like episodes, thyroid disease (subclinical or mild overt), brain MRI, absence of other neuronal antibodies in the serum or CSF, exclusion of alternative causes of encephalopathy by differential diagnosis, presence of antithyroid antibodies and responsiveness to immunotherapy. 

It is of note that there is no specific parameter for Hashimoto’s encephalopathy. Our analysis shows that anti-TPO may be a valuable diagnostic parameter for HE. Nonetheless, it is impossible to conclusively state that a patient with a positive anti-TPO result suffers from HE. Both patients with Hashimoto’s disease and those with Graves’ disease can present high anti-TPO levels, and thus this parameter cannot be effectively applied in the differential diagnosis. Additionally, anti-NAE was also suggested as a parameter indicating HE, yet it has been proven not to be specific only to HE.

Moreover, HE has a non-specific manifestation that most commonly involves disturbance of consciousness, impaired memory and sleep disturbance. Hence, HE should be considered in the differential diagnosis in patients with psychiatric and neurological symptoms, particularly when the symptom onset is sudden. Taking into consideration all the symptoms and clinical manifestations, HE should always be differentiated from other autoimmune encephalopathies and paraneoplastic syndromes, and the treatment should be introduced as soon as possible. 

The authors would like to emphasize the significance of conducting further research, which should aim to identify a specific parameter underlying HE, as well as performing randomized trials on large populations. Furthermore, at present, there are no studies indicating a genetic basis for the disease, although it is highly plausible that such mechanisms contribute to the development of HE.

Since HE may also co-exist with other autoimmune diseases, it is of the essence to gain a deeper understanding with regard to the underlying processes involved in the development of this disease.

## Figures and Tables

**Figure 1 ijms-25-07101-f001:**
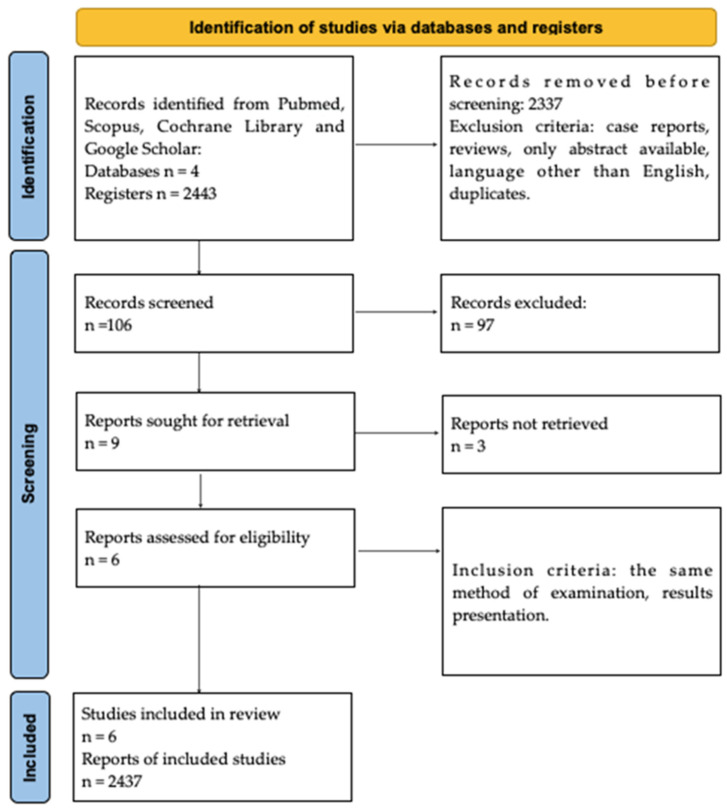
PRISMA Flowchart.

**Figure 2 ijms-25-07101-f002:**
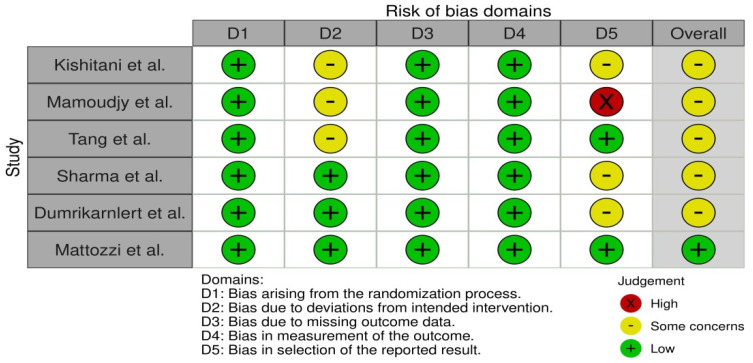
Bias assessment using RoB 2 tool [14,15,16,17,18,19].

**Figure 3 ijms-25-07101-f003:**
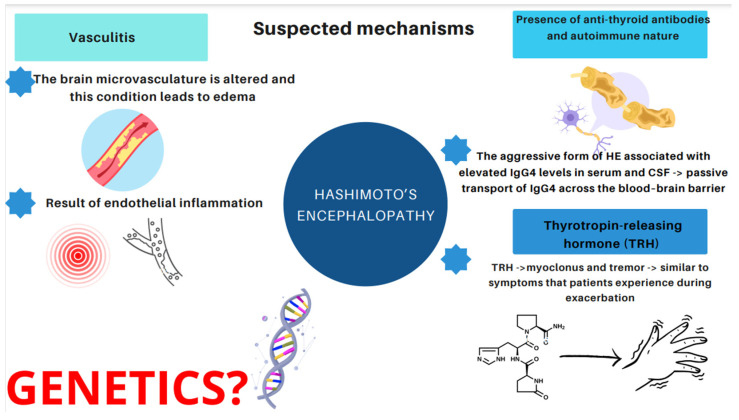
Suspected and proposed mechanisms of HE.

**Figure 4 ijms-25-07101-f004:**
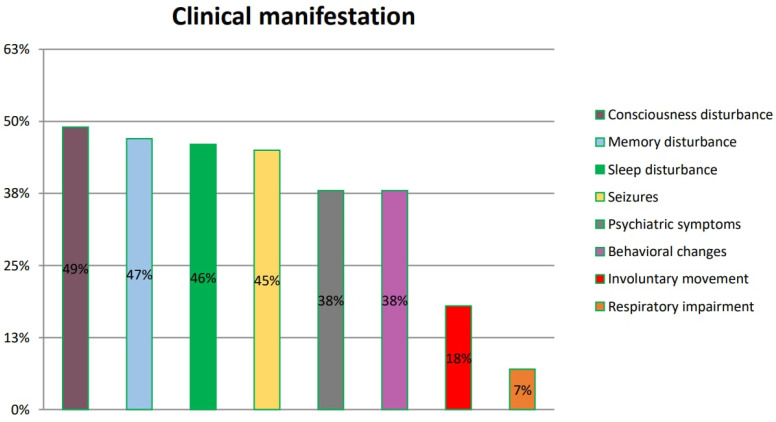
Symptoms of HE according to references. The symptoms listed in the figure were reported by patients and included in six original papers.

**Figure 5 ijms-25-07101-f005:**
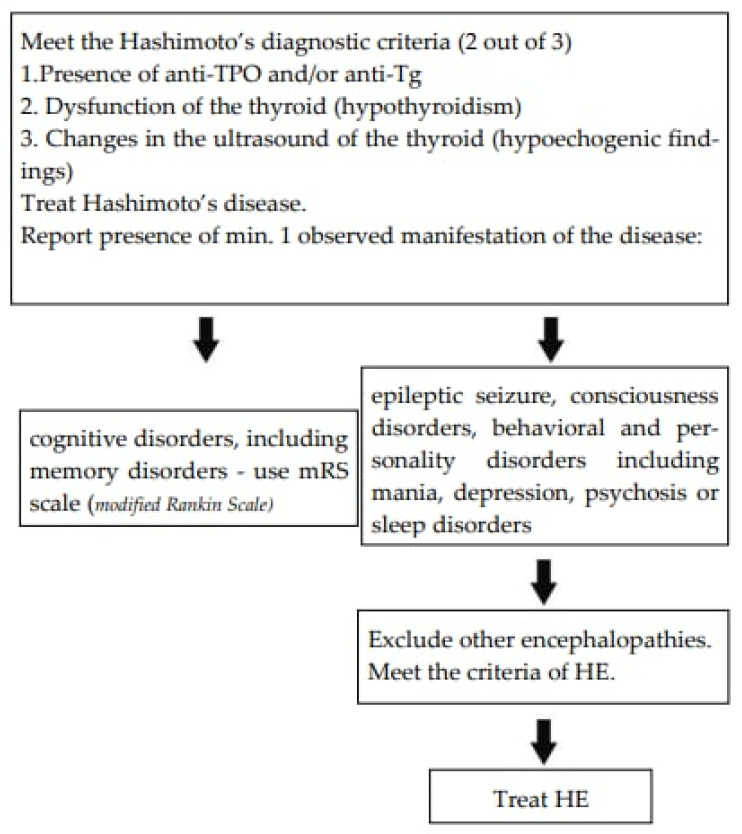
The suggested assessment scale [50].

**Table 1 ijms-25-07101-t001:** Clinical criteria of HE diagnosis according to the sources.

	Kishitani et al. [14]	Mamoudjy et al. [15]	Tang et al. [16]	Manjunatha Suryanarayana Sharma et al. [17]	Dumrikarnlert et al. [18]	Mattozzi et al. [19]
Encephalopathy with seizures, myoclonus, hallucinations or stroke-like episodes	**-**	**-**	**+**	**+**	**+**	**+**
Thyroid dysfunction (subclinical or mild dysfunction)	**-**	**+**	**-**	**-**	**+**	**+** (euthyroid status or mild hypothyroidism)
Brain MRI (Magnetic Resonance Imaging) normal or with nonspecific abnormalities	**-**	**-**	**-**	**-**	**+**	**-**
Absence of other neuronal antibodies in the serum or CSF (cerebrospinal fluid)	**-**	**-**	**-**	**-**	**+**	**+**
Exclusion of alternative causes of encephalopathy by differential diagnosis	**-**	**-**	**+**	**+**	**+**	**+**
Presence of antithyroid antibodies	**+**	**+**	**+**	**+**	**+**	**+**
Responsiveness to immunotherapy	**+**	**-**	**-**	**+**	**-**	**-**
Study group	14	8	13	13	13	24
Control group	26	34	No	No	91	13

**Table 2 ijms-25-07101-t002:** Laboratory findings according to sources.

	Kishitani et al. [14]	Mamoudjy et al. [15]	Tang et al. [16]	Manjunatha Suryanarayana Sharma et al. [17]	Dumrikarnlert et al. [18]	Mattozzi et al. [19]
Anti-TPO pre-treatment	No results	4043.3 ± 2969.8 IU/mL (8/8)	Elevated 13/13	Elevated 13/13, median range 909 IU/mL	Elevated 9/13	Elevated 24/24
Anti-NAE pretreatment	320–40.960	No results	No results	No results	No results	Elevated in 1 of suspected 24 HE and 1 of controls
Hyponatremia	6/14	No results	No results	No results	No results	No results
Elevated protein level in CSF	9/13	5/8	8/13	8/13	6/13	6/24
Abnormalities in EEG	11/12	8/8	7/13	7/13	No results	20/24
Abnormal TSH level	No results	2/8	0/13	5/13	No results	No results
Abnormal fT4 level	1/14	1/8	No results	No results	No results	No results
Elevated level of aminotransferase	No results	No results	5/13	3/13	No results	No results
ANA positive	No results	No results	2/7	4/13	2/6	No results
Rheumatoid factor positive	No results	No results	1/13	2/13	No results	No results
Elevated CRP level	No results	No results	1/13	3/13	1/13	No results

**Table 3 ijms-25-07101-t003:** Sensitivity and specificity of anti-thyroid peroxidase (anti-TPO) in the diagnostic process. Sensitivity is the percentage of persons with the disease who are correctly identified by the test. Specificity is the percentage of individuals without the disease who are correctly excluded by the test. Positive predictive value is the ratio of patients truly diagnosed as positive to all those who had positive test results. The negative predictive value is the proportion of the cases giving negative test results who are already healthy.

	Present Illness	Absent Illness	In Total
Positive patients	24	28	52
Negative patients	4	43	47
In total	28	71	

**Table 4 ijms-25-07101-t004:** Treatment according to sources.

Authors	Treatment			Time of Reatment	Steroid Therapy	mRS Score before Treatment	mRS Score after Treatment	Recovery after Steroids	Relapse after Steroids
	Drug	Patients	Dose						
Kishitani et al. [23]	methylprednisolone pulse therapy	13/14	1000 mg/day	3 days	13/14	(4.2 ± 0.9) mean ± SD *p* < 0.005	(1.7 ± 1.7) mean ± SD *p* < 0.005	11/13	3/13
	methylprednisolone pulse therapy followed by oral prednisolone therapy	9/14	median starting dose of prednisolone equal to 50 mg/d	No results					
	plasmapheresis	1/14	-	-					
Mamoudjy et al. [14]	i.v. methylprednisolone	2/8	30 mg/kg	3–5 days	5/8	No results	No results	No results	5/8
	p.o. prednisone	2/8	1 mg/kg/day	4 months					
	i.v. immunoglobulins	1/8	400 mg/kg/day	5 days					
Tang et al. [26]	corticosteroid therapy				8/13	No results	No results	5/8	1/8
Manjunatha Suryanarayana Sharma et al. [27]	i.v. methylprednisolone	10/13	1 g/d	5 days	12/13	No results	No results	8/12	2/12
	9 patients additionally p.o. steroids	9/13	No results	No results					
	prednisone 1 mg/kg/d	2/13	1 mg/kg/d	No results					
Dumrikarnlert et al. [28]	corticosteroid treatment.			No results	12/13	No results	No results	9/12	No results
Mattozzi et al. [29]	Steroids.			No results	19/24	No results	No results	6/19	8/19
Summary				69/85 (81.18%)				39/64 (60.94%)	19/60 (31.67%)

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
