# Peer review of "Difficulties in the Diagnostics and Treatment of Hashimoto’s Encephalopathy—A Systematic and Critical Review"

_ijms, 2024, doi:10.3390/ijms25137101_

Round 1
Reviewer 1 Report
Comments and Suggestions for Authors
Pempera and colleagues performed a systematic review regarding Hashimoto’s encephalopathy, a poorly understood disease entity. Methods are clear. However, I have major concerns regarding English language, results presentations, and manuscript structure. The logic is hard to follow, and the article is not easy to read.
- Clinical criteria are discussed in pathogenesis paragraph. Pathogenesis discussion is not the aim of this review (as stated) so I suggest to move clinical criteria discussion in another paragraph.
- The paragraph title “Research findings” is not specific. I suggest combining this paragraph with the clinical criteria discussion (Table 1) and change the title (i.e. “diagnostic criteria”);
- I can’t understand the aim and the meaning of the paragraph “What unites, what divides”. The author mentioned some of the limitations of the previous study, briefly discuss possible genetic basis and present laboratory findings without a clear logic. Please integrate this paragraph or eliminate it if not necessary for the general comprehension. Laboratory findings table (Table 2) can be moved in the following paragraph where they are already discussed;
- “Symptoms of HE according to references” paragraph is redundant. I suggest presenting these findings in a table or find a better way to present them being less redundant;
- “Consciousness” is not a symptom, it should be clarified what it means (consciousness alteration? coma?). The majority of the reported manifestation are “clinical manifestation”, not symptoms, so I suggest not to use the term symptom. Since the reports discussed are not too many more date regarding the combination of clinical manifestations could be discussed;
- No data regarding the prognosis are presented; please add and describe them. These could be very useful for the reader. Are specific clinical manifestation, laboratory findings, type and time of treatment associated to a worse or better prognosis?
- Some abbrevation’ explanations are lacking (i.e. mRS); please add a abbreviation paragraph at the end of the manuscript;
- Discussion pararaph is intereting in presenting some open questions in the field. I suggest moving 4.1 and 4.2 sections in the pathophysiology/pathogenesis paragraph (see before); some of the discussion anticipated in the previous text can be moved here to make the reading easier;
- I suggest a deep English revision from a native English speaker to make the article more understandable for the reader;
- Figure 1 can be better illustrated in a more captivating flowchart (arrows, colors, text centered in the boxes); Moreover, there are typos to correct;
- in Figure 2 the text under the figure is overlapping;
- Figure 4 does not have a legend, it is incomplete and data not understandable.
- In table 3 treatment column is difficult to understand. It could be useful to create a sub-structure (i.e. drugs, dosage, administration route etc);
- Given the target of the journal I would suggest add an additional figure concerning the pathogenesis which could better illustrate the complexity of the topic and the open questions;
Comments on the Quality of English Language
I suggest a deep English revision from a native English speaker to make the article more understandable for the reader.
Author Response
Dear Reviewer,
The authors thank you very much for the effort put into the review and for valuable comments. The comments helped us significantly improve the article.
We made corrections to the manuscript based on the comments. Each comment was considered separately and appropriate changes were made to our manuscript. The article has been revised by the expert of English language.
Yours Sincerely,
Authors

Reviewer 2 Report
Comments and Suggestions for Authors
In this review, the authors have summarized and analyzed the clinical reports about HE. The work seems to be interesting, and helpful for the clinical practice. I have some comments for the authors.
1.The authors mentioned that “A positive correlation was found between the level of thyroid peroxidase antibodies (anti-TPO) and thyroid fibrosis‘’. The authors need give the references or evidence for that.
2.In Page 7,“Presence of HE was correlated to presence of HE in Figure 3”. What does it mean? In the result section, “the sensitivity of anti-TPO is 85.714% and the specificity is 60.563%. 3.Therefore, it can be concluded that the positive predictive value is 46.154% and the negative predictive value is 91.489%”. The author need clarify the criteria for the anti-TPO positivity. Have they analyzed the predictive value of anti-TPO stratified by its serum level.
4.The authors propose that anti-NAE shouldn’t be used as diagnostic mean, since the findings were not consistent between T. Kishitani et al. and Mattozzi et al. And the authors also proposed that Hashimoto's encephalopathy is a diverse disease that differs in all the patients. So the author could not make an objective conclusion as above, and the autoantibodies against some epitopes in NAE may be a useful marker for part of HE patients, which is worth further exploration. The pathogenesis may be different between memory disturbance, and seizures presented by HE.
5. In Fig4, the authors need clearly label the symptoms presented by the HE patients.
6.Based on this systemic review, could the authors propose any new opinions in the diagnostic and therapeutic strategies for HE.
7. HE is a diverse and complex disease. It is still not very clear about its pathogenesis. Would the authors propose some suggestion about HE pathogenesis for the future studies?
Comments on the Quality of English LanguageThe gammar need minor modification in some places in this manuscript.
Author Response
Dear Reviewer,
The authors thank you very much for the effort put into the review and for valuable comments. The comments helped us significantly improve the article.
We made corrections to the manuscript based on the comments. Each comment was considered separately and appropriate changes were made to our manuscript. The article has been revised by the expert of English language.
Yours Sincerely,
